# Interactive effects of discharge reduction and fine sediments on stream biofilm metabolism

**Ana Victoria Pérez-Calpe**[1]*, **Aitor Larrañaga**[1], **Daniel von Schiller**[2], **Arturo Elosegi**[1]

**1** Department of Plant Biology and Ecology, University of the Basque Country (UPV/EHU), Bilbao, Spain,
**2** Department of Evolutionary Biology, Ecology and Environmental Science, University of Barcelona, Barcelona, Spain

* anavictoria.perez@ehu.eus

**Data Availability Statement:** All relevant data are within the paper and its Supporting Information files.

**Funding:** This research was funded by the Spanish Department of Economy, Industry and

## Abstract

Discharge reduction, as caused by water diversion for hydropower, and fine sediments deposition, are prevalent stressors that may affect multiple ecosystem functions in streams. Periphytic biofilms play a key role in stream ecosystem functioning and are potentially affected by these stressors and their interaction. We experimentally assessed the interactive effects of discharge and fine sediments on biofilm metabolism in artificial indoor channels using a factorial split-plot design with two explanatory variables: water discharge (20, 39, 62, 141 and 174 $cm^3 s^{-1}$) and fine sediments (no sediment or 1100 $mg L^{-1}$ of sediments). We incubated artificial tiles for 25 days in an unpolluted stream to allow biofilm colonization, and then placed them into the indoor channels for acclimation for 18 days. Subsequently, we manipulated water discharge and fine sediments and, after 17 days, we measured biofilm chlorophyll-a concentration and metabolism. Water velocity (range, 0.5 to 3.0 $cm s^{-1}$) and sediment deposition (range, 6.1 to 16.6 $mg cm^{-2}$) increased with discharge, the latter showing that the effect of increased inputs prevailed over sloughing. In the no-sediment treatments, discharge did not affect biofilm metabolism, but reduced chlorophyll-a. Sediments, probably as a consequence of nutrients released, promoted metabolism of biofilm and chlorophyll-a, which became independent of water discharge. Our results indicate that pulses of fine sediments can promote biofilm algal biomass and metabolism, but show interactive effects with discharge. Although discharge reduction can affect the abundance of basal resources for food webs, its complex interactions with fine sediments make it difficult to forecast the extent and direction of the changes.

## Introduction

Stream ecosystems are affected by multiple anthropogenic stressors [1]. Among these, damming and water diversion stand out as detrimental activities for stream biological communities [2–4] and ecosystem functioning [5]. The number of water diversion schemes is rising in response to escalating water demands [6,7]. Stream discharge reduction caused by water diversion reduces the width of the wet channel [8], affects water chemistry [9], alters transport and deposition of sediments [10], and impacts multiple ecosystem functions such as leaf litter breakdown [11–13], nutrient retention [14] and stream metabolism [15].

Competitiveness through the project GL2016-77487-R (DIVERSION), the European Social Fund, the Basque Government (Consolidated Research Group IT951-16) and the Biscay Province Council (61/2015). AVPC carried out this work thanks to a pre-doctoral grant by the Spanish Department of Economy, Industry and Competitiveness (BES-2017-081959). The funders had no role in study design, data collection and analysis, decision to publish, or preparation of the manuscript.

**Competing interests:** The authors have declared that no competing interests exist.

Fine sediments are also considered an important stressor and often included among the most prevalent pollutants in streams [16]. High inputs of fine sediments can occur as a consequence of natural processes [17,18], but are often exacerbated by human activities such as forestry or agriculture [19]. Suspended fine sediments reduce the light that reaches the stream bottom [20], can abrade biofilms [21], damage organisms gills [22,23] and interact with dissolved nutrients and other pollutants [24–26]. Additionally, fine sediments tend to settle on stream beds, where they cause siltation [27], reduce the supply of oxygen and light to the bottom and damage primary producers [28], macroinvertebrates [29] and fish [30].

Periphytic biofilms (hereafter biofilms) consist of complex communities of microorganisms that include bacteria, algae, fungi and protozoa, and live attached to rocks or other surfaces [31]. They play a key role in stream ecosystem functioning [32] and are an important food resource for invertebrates and fish [33]. The abundance, composition and activity of biofilms is regulated by factors such as light, current, nutrients and grazing [34]. Therefore, biofilms are highly sensitive to environmental changes and can be potentially affected by multiple anthropogenic stressors [35,36].

The response of biofilm to discharge reduction and fine sediments deposition is complex. In fast-flowing streams, water diversion reduces water velocity and shear stress, thus promoting biofilm growth and activity [37–39]. When natural discharge is low, further reductions can detrimentally affect biofilm by reducing nutrient exchange [34,40]. Besides, water diversion reduces the amount of fine sediments entering at reach, as most sediments are diverted with the water. At the same time, however, discharge reduction promotes the deposition rate of those sediments in the reach as a consequence of reduced water velocity, thus impacting benthic biota [41]. The final outcome will depend on factors such as water velocity, the characteristics of fine sediments or the type of organisms. Biofilms can be damaged by sediments via abrasion or burial [21], but can also benefit from fine sediments as a source of nutrients, especially phosphorus [42]. These complex interactions call for controlled experiments to examine how discharge and fine sediments affect biofilm structure and functioning.Here, we experimentally assessed the interactive effects of discharge and fine sediments on biofilm algal biomass and metabolism. The experiment was carried out in artificial stream channels, which were subject to a gradient of water discharge in presence or absence of fine sediments. We tested the following three hypotheses:

i. Algal biomass and metabolism will be lower at low discharge because of limited nutrient exchange.

ii. Addition of fine sediments will reduce algal biomass and metabolism because it hinders algal attachment and limits light availability.

iii. Water discharge and fine sediments will interact, algal biomass and metabolism being lowest in the channels with sediments and lowest water discharge.

## Materials and methods

### Experimental design

The indoor artificial stream facility of the University of the Basque Country (Leioa, Spain) consists of 30 indoor methacrylate channels (length-width-depth: 200-15-20 cm) grouped in six blocks of five channels. From a primary tank, filtered (1 mm mesh) rainwater is fed to six 200-L tanks (hereafter 'block tanks') that supply water to each block of five channels. In each block, water was recirculated by a pump and run as a closed system (Fig 1). Discharge can be adjusted for each channel individually. In each channel, water depth was set at 3.4 ± 0.1 cm

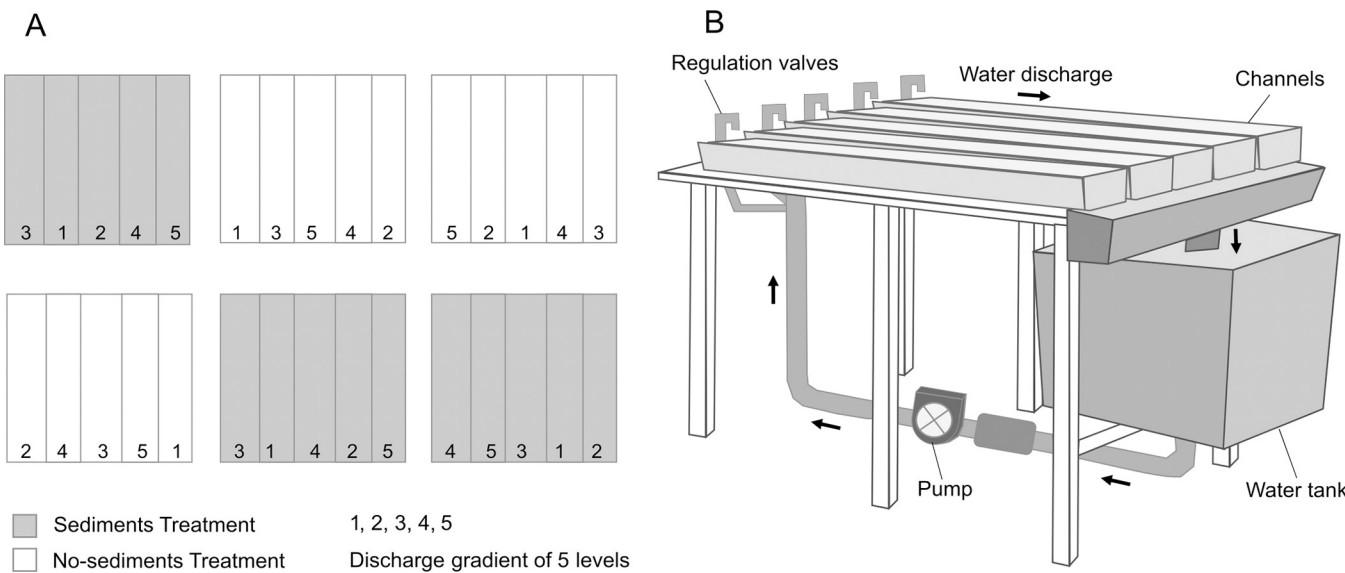

**Fig 1. Experimental setup.** (A) Schematic drawing of channel blocks setup and experimental levels of the factors discharge and fine sediments. (B) Detail of a channel block.

(mean ± SE) by means of a small dam at the lowermost end. LED lights (36 W 65000k, Aquael, Poland) with a 12/12 light/dark cycle and an intensity of 27.1 ± 1.0 µmol m$^{-2}$ s$^{-1}$ provided lighting. The bottom of the channels was covered by a 2-cm layer of commercial aquarium gravel of 8–16 mm average size (Karlie Flamingo, Germany). We used marble tiles (33.6 cm$^2$ of surface area) as standard biofilm substrata. To allow biofilm colonization, these tiles were incubated in an unpolluted and oligotrophic reach of the Urumea River (N Iberian Peninsula; 43˚ 12'40.6" N, 1˚54'06.2" W), attached on plastic trays with no protection from grazers and tied to the river bottom for 25 days.

After the incubation period, tiles were collected, transported to the artificial stream facility, and randomly distributed across the channels (12 tiles per channel). Additionally, to ensure biofilm development, we scraped several cobbles at the Urumea River, and the slush produced was split and uniformly distributed among the artificial channels. To allow biofilm acclimation, discharge was kept constant (discharge = 85.2 ± 2.48 cm$^3$ s$^{-1}$; water velocity = 1.7 ± 0.06 cm s$^{-1}$) in all the channels for 18 days. To avoid nutrient depletion, the water in each block tank was renewed every week during the acclimation and the experimental period.

After the acclimation period, we started a factorial experiment with a split-plot design, which lasted for 17 days. Water discharge was set in five levels (19.8 ± 1.3, 39.0 ± 3.8, 62.4 ± 5.1, 141.3 ± 8.6 and 173.7 ± 7.1 cm$^3$ s$^{-1}$) and fine sediments in two levels: no fine sediments and 1100 mg L$^{-1}$ of fine sediments, a concentration that is commonly found in the Urumea River during floods [43], as well as during forestry operations. Water discharge levels were randomly assigned to each channel within each block. Water discharge was measured on days 1, 10 and 14, from the time to fill a container at the lower end of each channel; water velocity was estimated from the ratio between water discharge and average channel width and depth, measured with a ruler along each channel. Water discharge and velocity remained constant during the experiment. Sediments were added to three randomly selected block tanks in 2 pulses (days 1 and 10). Sediments were distributed through the water pump and circulated through the treatment channels, where they settled rapidly, turbidity returning to background values a few hours after the addition.

The fine sediments used in this experiment were obtained from the recently emptied Enobieta Reservoir (43°12'50.5" N 1°47'31.0" W), located in the Urumea basin upstream from the biofilm collection point. These sediments were dried, ground and sieved through 200 µm. Their organic matter content was $21.0 \pm 0.11\%$ and their C:N molar ratio 49:1. Sediment leachate was characterized in the laboratory by mixing $2.0 \pm 0.01$ gr (n = 5) of dried sediments with 0.2 L of deionized water and kept at 20°C with light (120 µmol $cm^{-2}$) and with a constant movement (70 rpm in an orbital shaker Multitron II, INFORS HT, Bottmingen, Switzerland) for 24 h to mimic channel conditions. This leachate had a content of $0.093 \pm 0.003$ mg $g^{-1}$ of ammonium (N-$NH_4^+$), $0.012 \pm 0.0002$ mg $g^{-1}$ of nitrate (N-$NO_3^-$), $0.36 \pm 0.003$ mg $g^{-1}$ of total dissolved nitrogen (TDN), $3.96 \pm 0.001$ mg $g^{-1}$ of dissolved organic carbon (DOC) and $0.008 \pm 0.001$ mg $g^{-1}$ of soluble reactive phosphorus (SRP). Thus, sediment leachate contributed $20.9 \pm 0.2$ mg of N (sum of $NH_4^+$ and $NO_3^-$) and $1.7 \pm 0.2$ mg of P (from SRP) to each 5-channel block per sediment pulse. These quantities correspond to concentrations of $1.06 \pm 0.03$ mg $L^{-1}$ of N and $0.08 \pm 0.01$ mg $L^{-1}$ of P. See next section for analytical methods.

## Water quality

Water quality was analysed six times: on the first day of the experimental period, before and after renewing water (days 7 and 14) and on the last day of the experiment. We measured temperature, pH, electrical conductivity and dissolved oxygen concentration and saturation in the block tanks with a hand-held probe (Multi 3630 IDS, WTW, Germany). Water samples were collected from the block tanks, filtered through 0.7-µm pore size glass fibre filter (Millipore GF/F, Ireland) and stored at -20°C until analysis. The concentration of nitrate (N-$NO_3^-$), sulphate ($SO_4^{2-}$) and chloride ($Cl^-$), was measured by capillary electrophoresis (Agilent CE, Agilent Technologies, USA) [44]. The concentration of soluble reactive phosphorus (SRP) (molybdate method [45]) and ammonium (N-$NH_4^+$) (salicylate method [46]) were determined colorimetrically on a UV-1800 UV–vis Spectrophotometer (Shimadzu, Shimadzu Corporation, Kyoto, Japan). Total dissolved nitrogen (TDN) and total dissolved organic carbon (DOC) were determined by catalytic oxidation on a Shimadzu TOC-$L_{CSH}$ analyser coupled to a TNM-L unit (Shimadzu, Shimadzu Corporation, Kyoto, Japan).

## Response variables

At the end of the experiment biofilm variables were measured on the tiles. We measured chlorophyll-a (chl-a) concentration as a proxy of algal biomass by in vivo fluorimetry (Bentho-Torch, bbe Moldaenke Gmbh, Germany) in six randomly selected tiles in each channel. BenthoTorch is a non-intrusive tool that quantifies the total algal biomass through the stimulation of cell pigments and the reading of red fluorescent light emitted [47]. We summed the values of chlorophyll for green algae, cyanobacteria and diatoms, thus calculating total chl-a concentration [48].

Biofilm metabolism was estimated in 0.21-L glass chambers hermetically closed without recirculation. We enclosed one tile per chamber (6 random replicates per channel, 3 incubated in light conditions, 3 in dark conditions), filled them with water from the corresponding tank and incubated them for 1 h submersed in the channel. After incubation, we measured dissolved oxygen using a portable fibre optic oxygen meter with a syringe-like probe (Microsensor NTH-PSt7 on Microx 4, PreSens, Germany) by inserting its needle through the hermetic membrane. Metabolism metrics (i.e., gross primary production GPP, community respiration CR and net community metabolism NCM) were calculated following Acuña et al. [49]. We also calculated gross primary production per unit of algal biomass (i.e., GPP/Chl-a) as a proxy of metabolic efficiency [50,51].

Finally, we quantified the total amount of sediments deposited in the channels throughout the experiment by washing all the substrate within a container and measuring the turbidity (NTU) of the homogenised solution with a hand-held turbidimeter (AQ4500 Aquafast IV, Thermo Scientific Orion, USA). Turbidity (NTU) was converted to sediment concentration (g $L^{-1}$) using an empirical equation (*sediment concentration* = 0.0036 * *turbidity* + 0.0971, $r^2$ = 0.99, p < 0.001) established in the laboratory by measuring of turbidity of several solutions with a known concentration of the fine sediments (0, 0.1, 0.2, 0.5, 1, 2 and 4 gr $L^{-1}$).

## Data analysis

We analysed the differences among treatments in chlorophyll-a concentration ($\mu g\ cm^{-2}$) and biofilm metabolism metrics (GPP, CR, NCM; mg $O_2$ $h^{-1}$ and GPP/Chl-a; mg $O_2$ mg chl-a$^{-1}$ h$^{-1}$) using Linear Mixed-Effects Models (LMEM) with REML (function lme, in R package nlme [52]). Sediments (Yes vs. No) was set as fixed factor, Water discharge as a continuous explanatory variable, and blocks and channels nested within blocks, as random factors. We included a variance structure (varIdent in the nlme function) in the models to account for the variance heterogeneity between levels of the factor Sediments. The significance of each source of variation was tested by means of ANOVA. Chlorophyll-a concentration and GPP/Chl-a were log-transformed to meet homoscedasticity. All analyses were performed using R software, v. 3.4.0 [53].

## Results

### Water quality

The values of water temperature (22.4 ± 0.2˚C) and pH (7.5 ± 0.1) were stable during the experiment, with no differences among levels of the treatments (Table 1). A small, but significant, increase in dissolved oxygen concentration and saturation was observed in the sediment treatment (8.7 ± 0.1 mg $O_2$ $L^{-1}$ and 100.3 ± 0.8%) with respect to the no-sediment treatment (8.5 ± 0.1 mg $O_2$ $L^{-1}$ and 98.4 ± 1.3%) (Table 1). Electrical conductivity was lower in the sediment treatment (73.9 ± 2.8 $\mu S\ cm^{-2}$) than in the no-sediment treatment (104.7 ± 14.3 $\mu S\ cm^{-2}$)

**Table 1. Water quality values for each sediment treatment during the experiment.**

| Variable | (unit) | No-sediment | Sediment | F-value | P-value |
|----------|--------|-------------|----------|---------|---------|
| T | (˚C) | 22.3 ± 0.2 | 22.3 ± 0.2 | 0.22 | 0.644 |
| pH | - | 7.5 ± 0.1 | 7.5 ± 0.1 | 0.70 | 0.410 |
| DO | (%) | 99.2 ± 1.3 | 100.9 ± 0.7 | 7.02 | **0.013** |
| DO | (mg $L^{-1}$) | 8.6 ± 0.1 | 8.7 ± 0.1 | 8.60 | **0.007** |
| EC | ($\mu S\ cm^{-1}$) | 105.8 ± 13.0 | 75.2 ± 2.8 | 20.30 | **<0.001** |
| SRP | ($\mu g\ P\ L^{-1}$) | 18.1 ± 2.9 | 18.4 ± 2.9 | 0.08 | 0.783 |
| $NO_3^-$ | (mg N $L^{-1}$) | 0.6 ± 0.06 | 0.4 ± 0.08 | 8.61 | **0.007** |
| $NH_4^+$ | ($\mu g\ N\ L^{-1}$) | 22.2 ± 0.7 | 13.2 ± 0.2 | 2.82 | 0.104 |
| TDN | (mg N $L^{-1}$) | 1.8 ± 0.1 | 1.6 ± 0.1 | 3.90 | 0.058 |
| DOC | (mg C $L^{-1}$) | 3.2 ± 0.2 | 3.6 ± 0.3 | 3.38 | 0.076 |
| $Cl^-$ | (mg $L^{-1}$) | 5.7 ± 0.1 | 5.7 ± 0.1 | 0.07 | 0.796 |
| $SO_4^{2-}$ | (mg $L^{-1}$) | 5.6 ± 0.4 | 3.9 ± 0.1 | 17.30 | **<0.001** |

T, temperature; DO, dissolved oxygen concentration and saturation; EC, electrical conductivity; SRP, soluble reactive phosphorus; $NH_4^+$, ammonium; $NO_3^-$, nitrate; TDN, total dissolved nitrogen; DOC, dissolved organic carbon; $Cl^-$, chloride and $SO_4^{2-}$, sulphate. Values shown are mean ± SE. P-values and F-values were obtained by ANOVA. Degrees of freedom are 1, 29 for all variables. Significant P-values are shown in bold.

(Table 1), suggesting potential sorption of dissolved ions by added sediments. Most measured solutes did not differ between sediment and no-sediment treatments (Table 1). However, $NO_3^-$ and $SO_4^{-2}$ were significantly lower in the sediment treatment than in the no-sediment treatment (1.6 ± 0.3 and 2.7 ± 0.3 mg L$^{-1}$ vs. 4.0 ± 0.1 and 5.3 ± 0.4 mg L$^{-1}$, respectively), whereas DOC concentration was significantly higher in the sediment treatment (3.9 ± 0.2 vs. 3.2 ± 0.2 mg L$^{-1}$). Water renewal affected water quality: temperature, pH, dissolved oxygen, $SO_4^{2-}$ and DOC decreased (an average of 0.5°C, 0.5, 7.7% and 0.5 mg O$_2$ L$^{-1}$, 1.1 mg L$^{-1}$, 1.5 mg C L$^{-1}$, respectively) and, SRP and $NO_3^-$ increased (an average of 17.5 μg P L$^{-1}$ and 0.2 mg N L$^{-1}$, respectively) The rest of parameters showed no changes with water (S1 Dataset). Note that these changes were caused by water renewal, not by sediments, which were added to the corresponding treatments in days 1 and 10.

## Hydraulics and sediments

Water discharge correlated with water velocity (Table 2), which ranged from 0.5 to 3.0 cm s$^{-1}$ (Fig 2A). Sediments and discharge had a significant effect on deposited sediments (Table 2, Fig 2B). The amount of sediments deposited in the sediment treatment (6.1 to 16.6 mg cm$^{-2}$) was higher than in the no-sediment treatment (1.7 to 2.3 mg cm$^{-2}$). Discharge affected sediment deposition only in the sediment treatment. Sediments deposited increased with discharge, as a consequence of higher mass of sediments entering the channels, since all channels in a block received the same concentration but different mass of sediments.

**Table 2. Results of the Linear Mixed-Effects Models (LMEM) with water discharge as continuous explanatory variable, sediments as fixed factor and chlorophyll-a (Chl-a), gross primary production (GPP), community respiration (CR), net community metabolism (NCM), gross primary production per unit of chlorophyll-a (GPP/Chl-a), water velocity and deposited sediments as response variables.**

| Variable | | d.f. | F-value | P-value | Sign of coef. |
|---|---|---|---|---|---|
| Water velocity | Discharge | 1, 22 | 254.22 | <**0.001** | + |
| | Sediments | 1, 4 | 4.08 | 0.113 | |
| | Discharge × Sediments | 1, 22 | 0.24 | 0.629 | |
| Deposited sediments | Discharge | 1, 4 | 401.11 | <**0.001** | + |
| | Sediments | 1, 22 | 8.15 | **0.009** | + |
| | Discharge × Sediments | 1, 22 | 60.39 | <**0.001** | + |
| Chl-a | Discharge | 1, 22 | 3.79 | 0.064 | |
| | Sediments | 1, 4 | 11.97 | **0.026** | + |
| | Discharge × Sediments | 1, 22 | 6.67 | **0.017** | + |
| GPP | Discharge | 1, 22 | 0.21 | 0.647 | |
| | Sediments | 1, 4 | 43.40 | **0.003** | + |
| | Discharge × Sediments | 1, 22 | 0.85 | 0.366 | |
| CR | Discharge | 1, 22 | 2.31 | 0.143 | |
| | Sediments | 1, 4 | 20.26 | **0.011** | + |
| | Discharge × Sediments | 1, 22 | 0.21 | 0.647 | |
| NCM | Discharge | 1, 22 | 1.09 | 0.306 | |
| | Sediments | 1, 4 | 40.93 | **0.003** | + |
| | Discharge × Sediments | 1, 22 | 0.47 | 0.499 | |
| GPP/Chl-a | Discharge | 1, 22 | 1.65 | 0.211 | |
| | Sediments | 1, 4 | 0.85 | 0.408 | |
| | Discharge × Sediments | 1, 22 | 5.90 | **0.023** | - |

P-values, F-values and degrees of freedom (d.f.) were obtained by ANOVA. Significant P-values of main and interaction effects are shown in bold. The sign of the coefficient is indicated when the source of the variation is significant.

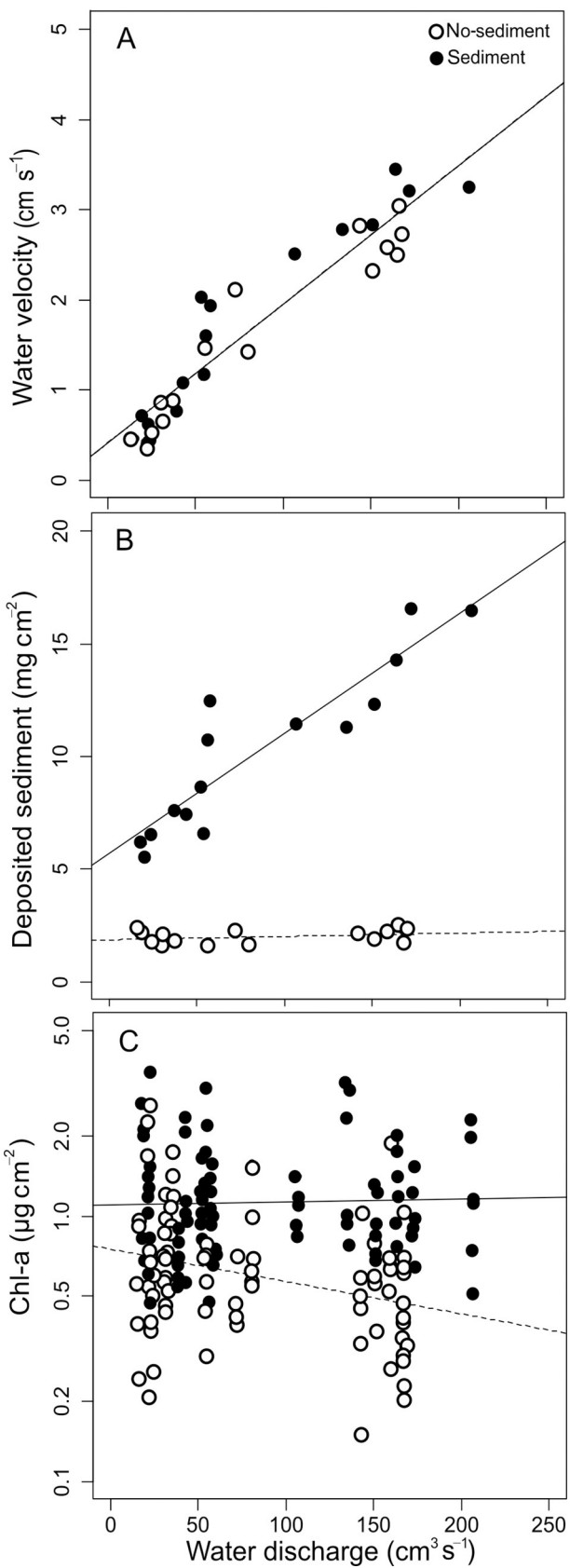

**Fig 2.** (A) Relationship between water discharge and velocity, (B) amount of deposited sediments in the channels and (C) chlorophyll-a concentration on tiles. Filled and empty dots correspond to channels with and without added sediments, respectively. Continuous and broken trend lines are built with the LMEM coefficients for channels with and without added sediments, respectively. When the interaction term is not significant a single line is shown. Note that in panels A and B each dot corresponds to a channel, and in panel C to a single tile.

## Biofilm

For the relationship between chl-a and discharge the LMEM showed a significant change of slope from the no-sediment to the sediment treatment (Table 2, p = 0.017). In the no-sediment treatment, chl-a concentration decreased significantly when water discharge increased, from $0.8 \pm 0.3$ µg cm$^{-2}$ in the channels with lowest discharge to $0.5 \pm 0.1$ µg cm$^{-2}$ in the channels with the highest discharge (Fig 2C). In the sediment treatment, on the other hand, chl-a concentration was higher ($1.3 \pm 0.1$ µg cm$^{-2}$) and constant along the discharge range. These results indicate that the sediments promoted biofilm chl-a and counteracted the negative effects of high discharge (Fig 2, Table 2).

The biofilm metabolism metrics did not change with discharge but increased significantly with the addition of fine sediments (Table 2, Fig 3). GPP rose from $41.6 \pm 6.3$ mg O$_2$ h$^{-1}$m$^{-2}$ in the no-sediment treatments to $92.1 \pm 11.9$ mg O$_2$ h$^{-1}$ m$^{-2}$ in the sediment treatments, CR from $13.2 \pm 2.0$ to $19.2 \pm 4.5$ mg O$_2$ h$^{-1}$ m$^{-2}$, and NCM from $28.4 \pm 5.6$ to $72.9 \pm 12.2$ mg O$_2$ h$^{-1}$ m$^{-2}$. The interaction between water discharge and the addition of fine sediments was not statistically significant for any metabolism metric. The GPP/Chl-a ratio showed no significant main effects of discharge or sediments; however, the significant interaction between both factors indicated that in the absence of sediments, increasing discharge resulted in a higher metabolic efficiency (p = 0.023, Table 2, Fig 4).

## Discussion

Our experiment assessed the interactive effects of water discharge and fine sediments on biofilm metabolism. We expected both, discharge reduction and sediments, to exert negative individual effects, as well as an interaction effect of both stressors, but these predictions were not supported by our results. Contrary to our expectations, both discharge reduction and fine sediments promoted biofilm biomass, their interaction resulting in unchanged biomass across all discharge levels in the sediment treatments. On the other hand, metabolism was positively affected by fine sediments, but unaffected by discharge.

According to our first hypothesis, we expected discharge reduction to negatively affect biofilm biomass and metabolism because of limited nutrient exchange. On the contrary, we observed a weak increase in chl-a and no changes in metabolism metrics with varying water discharge. The literature shows contrasting effects of water discharge on biofilms. Some studies reported no response for algal biomass [54–56] as well as for metabolism [14,54], whereas others showed that algal biomass decreased both above and below optimum velocities, a fact that would be explained by shear stress at high velocities, by nutrient limitation at low ones. This is the type of response reported by Biggs and Stokseth [57] where the algal biomass peaked at a velocity of 30 cm s$^{-1}$. Similarly, in a flume experiment, Hondzo and Wang [58] reported that shear stress reduced biomass and photosynthetic activity above 15 cm s$^{-1}$, whereas Liu and Lau [38] reported optimum biofilm growth at 1.5 cm s$^{-1}$. The discrepancies among studies are large and probably caused by differences in experimental conditions. Our water velocities were in the lower range of those so far mentioned, with a range between 0.5 and 3 cm s$^{-1}$, but even so, we found an inverse relationship between chlorophyll and velocity. This effect could be explained by the fact that our biofilm was dominated by loose algal filaments, which are the

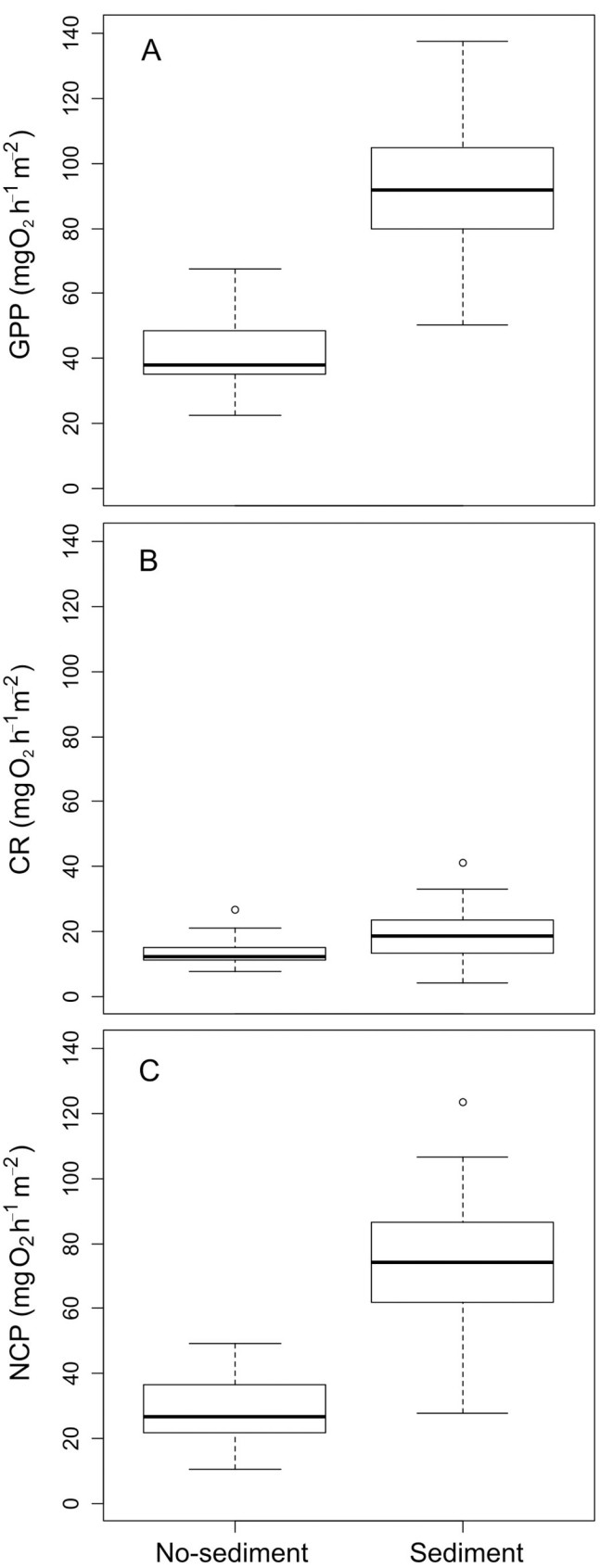

**Fig 3.** Differences in metabolism (A: Gross Primary Production, B: Community respiration and C: Net Community Metabolism) between channels with and without added sediments. The boxes display first and third quartiles, thick lines are medians, whiskers are range, and open circles are outliers.

growth forms dominant at low flow velocities [59]. The long and loose filaments in our experiment seemed especially prone to sloughing.

Our second hypothesis predicted that fine sediments would reduce biofilm biomass and metabolic activity, but we observed the opposite effect. The literature shows contrasting effects of fine sediments on biofilm. Several studies showed [41,60–62] fine sediments to reduce biofilm biomass and metabolism, but some [65,66] reported increased biofilm, which was explained as a consequence of shifts in the dominant growth forms towards those (e.g., motile algae) more resistant to physical disturbance. We did not study algal composition of biofilms in our experimental channels, but unlike biomass, which showed clear differences between sediment and no-sediment treatments, by the end of the experiment we did not see any visual difference in the appearance of biofilm. Alternatively, the effects of sediments on biofilm could be caused by nutrients, as their leachates had high concentrations of N and P, important nutrients for algae [42,63,64]. This fertilisation effect would, nevertheless, not depend strictly on the amount of sediments deposited in each channel, since the sediments were added into the block tank and dissolved nutrients from leachates would be distributed across all the channels with

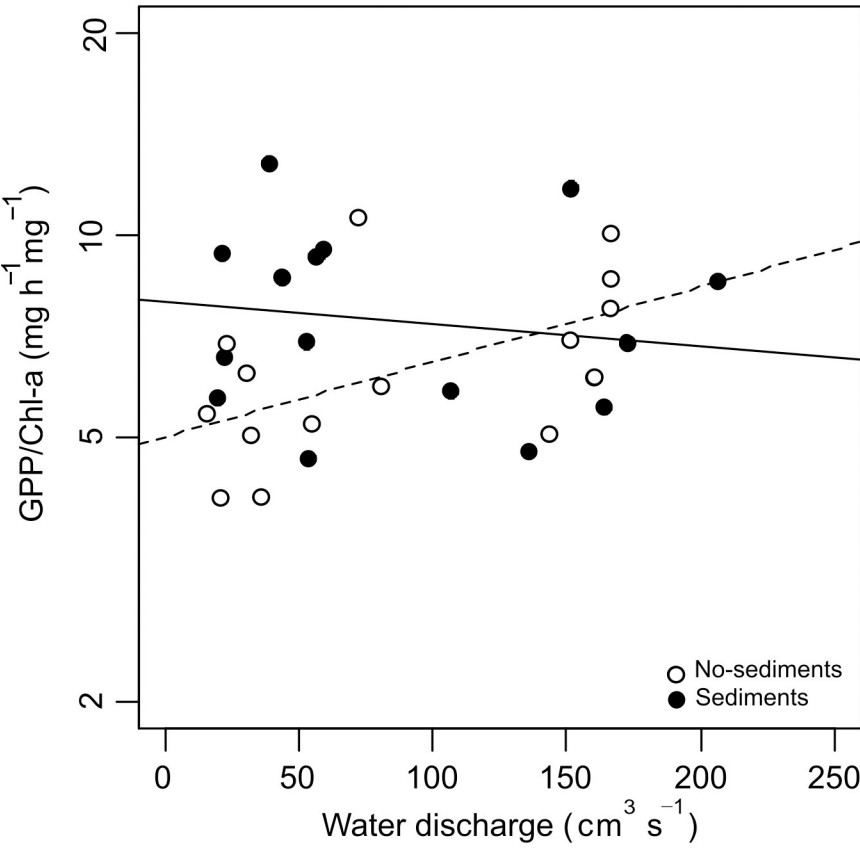

**Fig 4. Gross primary production per unit biofilm biomass (GPP/Chl-a).** Filled and empty dots correspond to channels with and without added sediments, respectively. The interaction between discharge and sediments is significant. Continuous and broken trend lines made with the LMEM coefficients for channels with and without added sediments, respectively.

the same concentration. The leaching of nutrients will of course depend on the type of sediments. The one used in our experiment, coming from a reservoir, could be unusually high in nutrients, but many other sediments also will act as fertilisers, as they are often linked to agricultural practices [17,19].

Our third hypothesis predicted algal biomass and metabolism to be lowest in the sediment treatments with lowest discharge. However, although our results showed a significant interaction, it consisted of sediments eliminating the effect of discharge on biomass. This response is consistent with a fertilizing effect that is stronger than the sloughing effect, at least under the experimental conditions. The effects of nutrients and flow velocity on algal biomass tend to interact in complex ways, flow velocity promoting turbulence and the diffusion of nutrients into biofilms [40], until shear stress increases so as to produce algal sloughing. In a recent study, Baattrup-Pedersen et al. [65] measured the metabolic and biomass response of periphytic biofilm to fine sediments, nutrients and discharge reduction and, although they found negative effects of sediments on chl-a concentration and GPP, they concluded that after 1 week, nutrient enrichment to some extent mitigated these negative effects. The metabolic efficiency for GPP showed a significant response to the interaction of discharge and sediments. At high discharge, it was similar in sediment and no-sediment treatments, whereas at low discharge it was higher in the sediment treatment. This difference could be probably explained by a greater stimulatory effect of sediment on GPP than on chl-a. At low discharges and sediment treatment, the positive effect of sediment would compensated any sediment-shading effect, whereas, with no-sediment treatment, the low efficiency with a high chl-a would result from self-shading of algal biomass [63,66]. Then, higher discharges would reduce biomass, but the remaining algae would be more efficient in absence of sediments. However, with higher sediment deposition the stimulatory effect would be overtaken by the sediment-shading effect that would reduce the metabolic efficiency [66] matching to the no-sediment metabolic efficiency.

In conclusion, discharge reduction and sediment inputs can have interactive effects on stream biofilm biomass and metabolism. Nonetheless, the direction and magnitude of the responses may be strongly site-specific and difficult to forecast, as they likely depend on the range of water velocities, on the composition of the fine sediments, as well as on the composition and biomass of benthic biofilms.

## Supporting information

**S1 Dataset. Excel spreadsheet containing the underlying numerical data for all figures and tables.**
(XLSX)

## Acknowledgments

The authors thank Ioar de Guzman, Miren Atristain and Maite Arroita for assistance in experimental and sampling work, and to SCAB from SGIKER (UPV/EHU/ERDF, EU) for technical and human support provided. We also acknowledge two unknown reviewers for their insightful comments.

## Author Contributions

**Conceptualization:** Aitor Larrañaga, Daniel von Schiller, Arturo Elosegi.

**Formal analysis:** Ana Victoria Pérez-Calpe, Aitor Larrañaga.

**Funding acquisition:** Aitor Larrañaga, Daniel von Schiller, Arturo Elosegi.

**Investigation:** Ana Victoria Pérez-Calpe, Aitor Larrañaga, Daniel von Schiller, Arturo Elosegi.

**Methodology:** Aitor Larrañaga, Daniel von Schiller, Arturo Elosegi.

**Resources:** Arturo Elosegi.

**Supervision:** Aitor Larrañaga, Daniel von Schiller, Arturo Elosegi.

**Visualization:** Ana Victoria Pérez-Calpe, Aitor Larrañaga.

**Writing – original draft:** Ana Victoria Pérez-Calpe, Aitor Larrañaga, Daniel von Schiller, Arturo Elosegi.

**Writing – review & editing:** Ana Victoria Pérez-Calpe, Aitor Larrañaga, Daniel von Schiller, Arturo Elosegi.

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
