## [Decision Letter · Decision Letter 0]

10 Sep 2020

PONE-D-20-22278

Interactive effects of flow reduction and fine sediments on stream biofilm metabolism

PLOS ONE

Dear Dr. Perez-Calpe,

Thank you for submitting your manuscript to PLOS ONE. After careful consideration, we feel that it has merit but does not fully meet PLOS ONE’s publication criteria as it currently stands. Therefore, we invite you to submit a revised version of the manuscript that addresses the points raised during the review process.

Both reviewers of your manuscript include comments on presentation style as well as substantive comments that deal with (for example) experimental design, statistical analysis, data interpretation, and your use and interpretation of the relevant literature. The comments on style are somewhat subjective, but please consider them carefully, since they are likely to improve the readability of your manuscript. The substantive comments are much less subjective, and you should pay particularly close attention to them. Also, pay special attention to the substantive comments of Reviewer 1, who is an expert on stream biofilms, hydrology, and sediment transport. If you disagree with some of the comments of either reviewer and therefore decide not to make the requested changes, be sure to justify your decision. This is especially important for substantive comments.

We look forward to receiving your revised manuscript.

Kind regards,

James N. McNair, Ph.D.

Academic Editor

PLOS ONE

Journal Requirements:

Reviewers' comments:

Reviewer's Responses to Questions

**Comments to the Author**

1. Is the manuscript technically sound, and do the data support the conclusions?

Reviewer #1: Partly

Reviewer #2: Partly

2. Has the statistical analysis been performed appropriately and rigorously? 

Reviewer #1: Yes

Reviewer #2: Yes

3. Have the authors made all data underlying the findings in their manuscript fully available?

Reviewer #1: Yes

Reviewer #2: Yes

4. Is the manuscript presented in an intelligible fashion and written in standard English?

Reviewer #1: Yes

Reviewer #2: Yes

5. Review Comments to the Author

Reviewer #1: This paper contributes to our understanding of sediment impacts on stream ecosystems, an understanding that is surprisingly limited given the magnitude of the issue. It is a strong paper in many ways: there are clearly stated hypotheses, a carefully designed well-replicated split plot experiment, an impressive test facility, and effective measurement of relevant variables. I have concerns, however, about how the results are interpreted in the context of previous work and of the limitations of the study design. The study found that sediments increase benthic biomass and productivity, and that (in the absence of added sediments) there is negative relationship between benthic biomass and water velocity. These are potentially useful results, but both supporting and contradictory results appear in the literature. The authors do cite results from both sides, and they acknowledge that, “[t]he response of biofilm to flow reduction and fine sediment deposition can be complex” (line 67). But their effort to resolve these complexities is inadequate. The authors further found that, in the presence of added sediments, their observed negative effect of velocity disappeared. This is perhaps the only truly novel result of the study yet their attempt to explain it (lines 296-298) falls short. I return to these concerns with more specifics in the comments that follow.

The term “flow reduction” is confusing in that a positive response to flow reduction is a negative response to flow. Further, the experiment involved both increases and decreases relative to the acclimatized flow. I would limit the use of “flow reduction” to the introduction and discussion.

Line 35: Unclear. How about “ranges of … and of” rather than “ranges from … and from”

Lines 69-72. Too many reversals: “Conversely…On the other hand…However”

Line 78. Missing here is a transition from the general problem to the specific design. What answers were being sought that motivated this particular experiment (these particular hypotheses)?

Line 95: change to “filtered (1 mm) rainwater”

Line 95: How fast was the water fed from the primary tank to the block tanks? Or did the system run closed for extended periods, renewed in batch on days 7 and 14 (line 137)? I wanted to see this to infer the mass of sediment actually added and to interpret the water quality data vis a vis water renewal.

Line 95: A diagram of the set-up (like Matthaei et al. 2010, Fig 1) would help a lot. I’ve had to assume that all recirculation passed through the block tank and that there were not separate overflows from each channel. I’ve just now seen that is likely the system ran closed.

Line 128: We need a conversion of the “lixiviates” (a new term for me) into mass of nutrients that were potentially available to the biofilms. E.g., the solid:liquid ratio represented by these concentrations, and then the mass of sediments transferred to the channels. This is necessary to evaluate the inference that the sediments stimulated the biofilm through nutrient release. More below.

Line 136: Apparent discrepancies: As the sentence reads, I count six, not four, sample times. There are five sets in the Supplementary Data. If the numbers under “Sampling” in the Supplement represent experiment days, these do not match the description on lines 136-137.

Line 171: The conversion of 0.0036 mg/(L-NTU) seems way too low. It usually runs between 1 and 2.

Line 177: Insert abbreviation “(LMEM)”

Lines 187-203 (Water Quality). The potential for nutrients to have stimulated benthic biomass raises the question of whether there was a temporal signal in the water quality data. It appears to me from the Supplemental Data that there was not, but perhaps the timing of the WQ samples was such that a short-term pulse would have been seen. This issue should be addressed in the text, if only to say that none was detected and perhaps why none was detected. The Supplemental Data should clearly identify the sampling date.

The data for sediment deposited in the Supplemental Data are ten times higher than in Figure 1B. Were these in g/m2 but mislabeled as mg/cm2?

Line 206: Strike “Results from the LMEM showed that”. LMEM is made clear in Table 2.

Line 207: Strike “LMEM determined that”

210: Did deposited sediment really increase with flow in the no-sediment treatment? The broken line in Fig. 1B is nearly horizontal. The R2=0.44 strikes me as impossible. I would guess 0.01.

211-213: Admitting I am not a statistician, I would say that the interaction simply verified that the flow effect on deposition differed by treatment.

Lines 214-221 (Table 2). Could be simplified. The F-value could be eliminated as (nearly) redundant on P-value. I see no value in the intercept statistics. Thanks for the d.f.s. They are important.

Line 231: As noted above, it would help to frame responses in terms of flow increase, rather than decrease.

Line 237: Replace “scouring” with something else. “negative”? I don’t think you were observing scouring; perhaps sloughing or detachment of strands.

line 243: Replace “rate” with “metric”, “parameter”, or something. “rate” is confusing here because it suggests that rates were quantities.

Lines 244-245: The regression lines in Figure 3 do not fit the data. I regressed the (log) relationships from the Supplemental Data. The no-sediment line has a significant (P~0.025) upward slope and an R2 of 0.35. The sediment line has a near-zero slope and an R2 of 0.01. This affects interpretation. Flow has no effect in the presence of sediments. Metabolic efficiency increases with flow in the absence of sediments. At low flows, metabolic efficiency is higher in the presence of sediments than in their absence, but the two are equal at high flows. Basically what we see here is (1) the inverse of the no-sediment chlorophyll-flow relationship (i.e., since chlorophyll declined but GPP didn’t, metabolic efficiency increased), together with (2) an enhancement of metabolic efficiency at all flows in the presence of sediment (GPP increased more than chlorophyll).

Lines 251,252: Strike “When” and end the sentence at “significant”.

Lines 259,260. I suggest: “biomass, and their interaction was not additive but antagonistic.” “Antagonistic” is a technical term in statistics but not all readers know this.

Line 255 (DISCUSSION). You need to address why sediment deposition increased with water flow. This result is presented in the abstract and shown prominently as Fig. 1 B. I was at first puzzled, thinking that more should deposit at the slower velocity, until I realized that the greater delivery at higher flows would have overridden this effect. The tight correlation between deposited sediment and water velocity is potentially important: it means that any response to water flow observed in sediment-treated mesocosms would have been indistinguishable from a response to the quantity of deposited sediment. Fortunately, there were no such responses.

Lines 265-267 (Discussion of velocity effects). First, some corrections. On line 270, “above” should read “to” or better yet, delete after “declined” because this referred to a different life form, one that had no optimum. Use consistent units, which here would be cm/s. Thus, Biggs’ optimum was 20 cm/s. Honzo and Wang’s optimum was 15 cm/s, not 0.7 cm/s which is a shear velocity not a water velocity.

I think this section should be rewritten to clarify. Point out that many studies have observed an optimum velocity, that these are typically in the range of 20 cm/s or more, but that your results suggest either an extremely low optimum (< 3 cm/s) or none at all, which is consistent with Lau and Liu, and with the result of Biggs et al. for long filamentous forms. You then need to speculate on why your result is unusual – were there long filamentous forms? You should not dismiss your result (“we did not observe any major effect”) by appealing to the small velocity range, because it is a central result of the study.

Lines 277-288 (Discussion of sediment effects). The unexpected result that sediments enhanced biofilm growth should be treated with care because it could be used to downplay the need to control sediment inputs. The claim that the result “is line with findings of other mesocosm studies” should be modified to acknowledge that it is not in line with most mesocosm studies. Of the two cited, Izagirre found that growth rates were below control for higher additions. I agree that for your case, it is probably nutrients that explain the increase, but the citation to Magbanua is wrong: they ruled out nutrients and went with Izagirre’s habitat argument. Pigotti attributed it to growth form. The evidence that you adduce for nutrients could be considerably strengthened by a quantitation of the nutrients actually added or by evidence from the water quality data. I lean in favor of nutrients because that explains why the stimulatory effect of sediments was entirely independent of the quantity deposited. Since all flumes were fed from a single tank, the nutrient exposure would have been the same across all flows and therefore across all levels of bed sediment. Finally, it should be pointed out that if nutrients were responsible, this is not a widely applicable result because most sediment inputs do not originate from lake beds.

Line 292 “but this effect was smaller at low flows” is perhaps technically correct but misses the obvious: that with sediment addition the negative effect of higher velocity disappeared. Framed in this latter way, the next sentence (lines 292-294), which I cannot understand, becomes unnecessary.

Line 297: So how might the sediments have increased the resistance of algae to higher flow? This question needs answering and I suggest the answer lies in the nutrient effect. Horner and Welch (1981) found that the optimum velocity for algal growth increased with increasing phosphorus concentration. Basically, at higher velocities, the higher nutrients allow the algae to grow faster than they are sheared. In your case, the added nutrients may have shifted the optimum upward enough to eliminate the negative velocity effect.

Line 299 In what sense? I’m confused.

Lines 308-313. The higher metabolic efficiency at low flows in the presence sediment, where there is more chlorophyll, cannot be related to self-shading if we take chlorophyll as an indicator of biomass. Under self-shading, the ratio of GPP to biomass goes down, not up. In the absence of sediment, the increase in metabolic efficiency with flow is associated with a loss of chlorophyll (biomass), so in this case, the trend is consistent with self-shading.

Reviewer #2: Review: Perez-Calpe et al. interactive effects of flow and fine sediments on stream biofilms

Overall I enjoyed reading this study, and only have one significant recommendation related to how the authors treat the concept of scouring flows in the context of their replicated stream channels. I think the author’s conclusions are valid and support what has previously been published, but am not convinced that there are significant new insights provided. They certainly make a case that the interaction between suspended and deposited sediments and their subsequent contribution to nutrient loading and periphyton metabolism is important to consider. E.g., not all sediment is created equal. They also provide evidence that the way in which sediment is deposited onto the biofilm is important, but it is a bit challenging to apply these results to the ‘real world’ because of some experimental design artifacts. Finally, I DO believe these concerns can be addressed and that the manuscript would be valuable to readers in this subdiscipline.

Abstract

Line 34: Why would lower water velocity lower sediment deposition? Doesn’t make sense to me. Faster water velocity should entrain more sediment and keep the tiles clearer of sediment deposition?

Paragraph ending with line 78: I really liked this summary of the varied and complex interactions that occur with biofilm metabolism as a result of flow modification. Excellent.

Line 99: change ‘was’ to ‘were’.

Line 103: were grazers kept from incubating tiles? Please clarify.

Line 110: Did you measure/quantify dissolved nutrients? Seven days of recycled water could lead to significant nutrient reduction although you may have quantified this.

Line 121: A couple of concerns here: in a ‘natural’ flood, sediment would remain in suspension for a longer period of time and abate during the falling limb of the hydrograph. By not resuspending the sediment additions, I’m not sure how much of an effect you would have, although this may become evident when looking at your AFDM or turbidity data (e.g., suspended sediment was deposited within the stream channels based on water velocity characteristics). Another concern with the sediment addition is related to dissolved and sediment-derived nutrients. As you noted in the introduction, sediment comes hand-in-hand with nutrients that can be released and utilized by the biofilm. So, how do you know that the results you obtained weren’t due to differences in dissolved nutrients between treatments? This is an important, potentially confounding factor in your study design. Perhaps you include this in your analyses.

Line 191: Interesting! Why do you think conductivity was lower in the sediment treatment? This doesn’t make sense to me unless the sediment was adsorbing dissolved ions.

Line 214: Table 2. Is there a way to indicate the direction the significant interactions? For example, deposited sediment increased with increasing flow, etc.

Line 236: Although I agree with your interpretation of the data—namely that sediment deposition on the tiles increased metabolism, I don’t think you can claim that higher flows resulted in scouring. If that were the case, sediment deposition on the tiles would be lower in the high velocity treatments. Right? I think you can fix this by changing the wording here, and would strongly advise that you restructure the materials/methods too. I’d refrain from talking about scouring flows, and instead simply note that sediment deposition in channels with higher flow and velocity was an intentional artifact of your sediment/no-sediment treatment. It seems to me that when you dosed your feeder tanks with sediment, there was more sediment deposited on tiles in channels with higher flows because they were pushing more sediment (mass per unit time) over the channel. Does this make sense? Obviously there is no scouring going on at these velocities or you wouldn’t see this pattern.

Line 244: change ‘showing’ to ‘showed’.

Line 245: I am looking at Figure 3, and although I appreciate your statistical inference, I’m wondering if you believe it is ecologically significant? E.g., is the pattern indicating an ecological response, or is it an artifact of the experimental design and indicative of significant variability in the data?

Line 276 and preceding paragraph: Excellent review of some of the key papers looking at biofilm/velocity relationships.

Line 317: Discussion—Very nice. In particular, I appreciated your thoughtful review of your initial hypotheses. I also appreciated your observation that velocities used in this study were on the low side relative to the literature (aka Biggs et al., etc.).

Minor point: I find that some readers mistake ‘flow’ for ‘velocity’. I’d recommend that you change this to ‘discharge’ throughout.

6. PLOS authors have the option to publish the peer review history of their article (what does this mean?). If published, this will include your full peer review and any attached files.

Reviewer #1: No

Reviewer #2: **Yes: **Eric B Snyder

---

## [Author Response · Author response to Decision Letter 0]

22 Oct 2020

Dear Dr. McNair,

A revised version of the manuscript PONE-D-20-22278 “Interactive effects of flow reduction and fine sediments on stream biofilm metabolism”, now entitled “Interactive effects of discharge reduction and fine sediments on stream biofilm metabolism” by Ana Victoria Pérez-Calpe, Aitor Larrañaga, Daniel von Schiller and Arturo Elosegi, is attached to your consideration for publishing as an article in PLOS ONE journal. 

We are very grateful to the reviewers for their detailed and positive comments. We have followed closely these comments, as we detail below. 

We hope that the present version of the Ms reaches the standards for PLOS ONE journal.

Best regards,

Ana Victoria Pérez-Calpe, PhD student

Faculty of Science and Technology

University of the Basque Country,

P.O. Box 644, 48080 Bilbao, Spain

Tel.: +34 94 601 5939

Email: anavictoria.perez@ehu.eus

 

Comments to the Author

Reviewer #1: This paper contributes to our understanding of sediment impacts on stream ecosystems, an understanding that is surprisingly limited given the magnitude of the issue. It is a strong paper in many ways: there are clearly stated hypotheses, a carefully designed well-replicated split plot experiment, an impressive test facility, and effective measurement of relevant variables. I have concerns, however, about how the results are interpreted in the context of previous work and of the limitations of the study design. The study found that sediments increase benthic biomass and productivity, and that (in the absence of added sediments) there is negative relationship between benthic biomass and water velocity. These are potentially useful results, but both supporting and contradictory results appear in the literature. The authors do cite results from both sides, and they acknowledge that, “[t]he response of biofilm to flow reduction and fine sediment deposition can be complex” (line 67). But their effort to resolve these complexities is inadequate. The authors further found that, in the presence of added sediments, their observed negative effect of velocity disappeared. This is perhaps the only truly novel result of the study yet their attempt to explain it (lines 296-298) falls short. I return to these concerns with more specifics in the comments that follow.

The term “flow reduction” is confusing in that a positive response to flow reduction is a negative response to flow. Further, the experiment involved both increases and decreases relative to the acclimatized flow. I would limit the use of “flow reduction” to the introduction and discussion.

Response: We followed the advice of the reviewer and now talk of "discharge" instead of "discharge reduction". The latter is only used in the Introduction, to frame the experiment in the context of alterations in discharge, especially reductions caused by water diversion.

Line 35: Unclear. How about “ranges of … and of” rather than “ranges from … and from”

R: Modified. Line: 37.

Lines 69-72. Too many reversals: “Conversely…On the other hand…However”

R: Modified. 

Line 78. Missing here is a transition from the general problem to the specific design. What answers were being sought that motivated this particular experiment (these particular hypotheses)?

R: Added in the text. Line 78. 

Line 95: change to “filtered (1 mm) rainwater”

R: Modified. Line 95.

Line 95: How fast was the water fed from the primary tank to the block tanks? Or did the system run closed for extended periods, renewed in batch on days 7 and 14 (line 137)? I wanted to see this to infer the mass of sediment actually added and to interpret the water quality data vis a vis water renewal.

R: Blocks ran as closed system for periods of 7 days, and for this reason, day 7 and 14 water was renewed. We have clarified this in the text. Line 97.

Line 95: A diagram of the set-up (like Matthaei et al. 2010, Fig 1) would help a lot. I’ve had to assume that all recirculation passed through the block tank and that there were not separate overflows from each channel. I’ve just now seen that is likely the system ran closed.

R: The reviewer is right that all channels in the same block received the water from the same tank. A diagram has been included as Fig 1. Line 97.

Line 128: We need a conversion of the “lixiviates” (a new term for me) into mass of nutrients that were potentially available to the biofilms. E.g., the solid:liquid ratio represented by these concentrations, and then the mass of sediments transferred to the channels. This is necessary to evaluate the inference that the sediments stimulated the biofilm through nutrient release. More below.

R: Sorry, it was a mistake. We have changed the term “lixiviate” to “leachate”. Moreover, we now provide the characterization of the leachates (mg/g of sediment) and the mass of nutrients they provided to each tank. We could not calculate the mass of nutrients transferred to each channel, since the fact that channels in the same block were all fed with water from the same tank determines that the dissolved nutrients were those leached from the total sediments in the block, not from the sediments retained in each channel. Line 135.

Line 136: Apparent discrepancies: As the sentence reads, I count six, not four, sample times. There are five sets in the Supplementary Data. If the numbers under “Sampling” in the Supplement represent experiment days, these do not match the description on lines 136-137.

R: Sorry, the reviewer is right, there were six samplings. We have corrected the text and the supplementary data and Table 1 has been updated. 

Line 171: The conversion of 0.0036 mg/(L-NTU) seems way too low. It usually runs between 1 and 2.

R: Sorry, there was a mistake in sediment concentration units. It has been corrected. Line 178.

Line 177: Insert abbreviation “(LMEM)” 

R: Done. 

Lines 187-203 (Water Quality). The potential for nutrients to have stimulated benthic biomass raises the question of whether there was a temporal signal in the water quality data. It appears to me from the Supplemental Data that there was not, but perhaps the timing of the WQ samples was such that a short-term pulse would have been seen. This issue should be addressed in the text, if only to say that none was detected and perhaps why none was detected. The Supplemental Data should clearly identify the sampling date.

The data for sediment deposited in the Supplemental Data are ten times higher than in Figure 1B. Were these in g/m2 but mislabeled as mg/cm2?

R: Water renewal caused some small changes in water quality. It has been added in text. Line 204. 

Sampling dates has been included in supplementary data and the mistake in the magnitude of sediment deposition has been corrected. 

Line 206: Strike “Results from the LMEM showed that”. LMEM is made clear in Table 2. 

R: Done.

Line 207: Strike “LMEM determined that” 

R: Done.

210: Did deposited sediment really increase with flow in the no-sediment treatment? The broken line in Fig. 1B is nearly horizontal. The R2=0.44 strikes me as impossible. I would guess 0.01.

R: Thanks to the referee comment, we noticed that showing R2 and ANOVA information could be confusing, and we decided to remove the R2 and only use the information provided by ANOVAs. Thus, Fig 1A and 1B (now renamed Fig 2A and 2B) have been modified in this sense. 

In Fig 1C (now Fig 2C) to avoid misunderstanding with solid and dotted lines of trend, we have eliminated them. 

211-213: Admitting I am not a statistician, I would say that the interaction simply verified that the flow effect on deposition differed by treatment.

R: Modified. Line 320.

Lines 214-221 (Table 2). Could be simplified. The F-value could be eliminated as (nearly) redundant on P-value. I see no value in the intercept statistics. Thanks for the d.f.s. They are important.

R: We eliminated the intercept. On the other hand, we find the F-value is necessary as it is a standard way to compare the variability among treatments with the variability within. Therefore, we prefer to maintain it. 

Line 231: As noted above, it would help to frame responses in terms of flow increase, rather than decrease.

R: Results section have been rewritten in terms of flow increase. 

Line 237: Replace “scouring” with something else. “negative”? I don’t think you were observing scouring; perhaps sloughing or detachment of strands.

R: Sorry, our mistake. We have replaced the term “scour” with “slough” in all the text. 

Line 243: Replace “rate” with “metric”, “parameter”, or something. “rate” is confusing here because it suggests that rates were quantities.

R: We have replaced “metabolic rate” with “metabolism metric” in all the text.

Lines 244-245: The regression lines in Figure 3 do not fit the data. I regressed the (log) relationships from the Supplemental Data. The no-sediment line has a significant (P~0.025) upward slope and an R2 of 0.35. The sediment line has a near-zero slope and an R2 of 0.01. This affects interpretation. Flow has no effect in the presence of sediments. Metabolic efficiency increases with flow in the absence of sediments. At low flows, metabolic efficiency is higher in the presence of sediments than in their absence, but the two are equal at high flows. Basically what we see here is (1) the inverse of the no-sediment chlorophyll-flow relationship (i.e., since chlorophyll declined but GPP didn’t, metabolic efficiency increased), together with (2) an enhancement of metabolic efficiency at all flows in the presence of sediment (GPP increased more than chlorophyll).

R: Thanks to the reviewer comment, we noticed that the dotted slope line was incorrect. We have modified the Fig 3 (now renamed Fig 4). Moreover, interpretation has been changed in the discussion section. Line 325.

Lines 251,252: Strike “When” and end the sentence at “significant”. 

R: Done. 

Lines 259,260. I suggest: “biomass, and their interaction was not additive but antagonistic.” “Antagonistic” is a technical term in statistics but not all readers know this. 

R: Text has been rewritten avoiding “antagonistic” and “synergistic” terms to improve understanding.

Line 255 (DISCUSSION). You need to address why sediment deposition increased with water flow. This result is presented in the abstract and shown prominently as Fig. 1 B. I was at first puzzled, thinking that more should deposit at the slower velocity, until I realized that the greater delivery at higher flows would have overridden this effect. The tight correlation between deposited sediment and water velocity is potentially important: it means that any response to water flow observed in sediment-treated mesocosms would have been indistinguishable from a response to the quantity of deposited sediment. Fortunately, there were no such responses.

R: Done. Lines 126 and 226.

Lines 265-267 (Discussion of velocity effects). First, some corrections. On line 270, “above” should read “to” or better yet, delete after “declined” because this referred to a different life form, one that had no optimum. Use consistent units, which here would be cm/s. Thus, Biggs’ optimum was 20 cm/s. Honzo and Wang’s optimum was 15 cm/s, not 0.7 cm/s which is a shear velocity not a water velocity. 

I think this section should be rewritten to clarify. Point out that many studies have observed an optimum velocity, that these are typically in the range of 20 cm/s or more, but that your results suggest either an extremely low optimum (< 3 cm/s) or none at all, which is consistent with Lau and Liu, and with the result of Biggs et al. for long filamentous forms. You then need to speculate on why your result is unusual – were there long filamentous forms? You should not dismiss your result (“we did not observe any major effect”) by appealing to the small velocity range, because it is a central result of the study.

R: Done. We have rewritten the section following the reviewer suggestions and we have focused the discussion on biofilm growth form. Lines 280-296.

Lines 277-288 (Discussion of sediment effects). The unexpected result that sediments enhanced biofilm growth should be treated with care because it could be used to downplay the need to control sediment inputs. The claim that the result “is line with findings of other mesocosm studies” should be modified to acknowledge that it is not in line with most mesocosm studies. Of the two cited, Izagirre found that growth rates were below control for higher additions. I agree that for your case, it is probably nutrients that explain the increase, but the citation to Magbanua is wrong: they ruled out nutrients and went with Izagirre’s habitat argument. Pigotti attributed it to growth form. The evidence that you adduce for nutrients could be considerably strengthened by a quantitation of the nutrients actually added or by evidence from the water quality data. I lean in favor of nutrients because that explains why the stimulatory effect of sediments was entirely independent of the quantity deposited. Since all flumes were fed from a single tank, the nutrient exposure would have been the same across all flows and therefore across all levels of bed sediment. Finally, it should be pointed out that if nutrients were responsible, this is not a widely applicable result because most sediment inputs do not originate from lake beds.

R: Paragraph rewritten. We have corrected the citations. Following the reviewer advice, we have strengthened the fertilisation effect including in methods section the amount of nutrients contained in sediments pulses and we have pointed the unusually nutrient concentration of our sediments. Lines 297-313. 

Line 292 “but this effect was smaller at low flows” is perhaps technically correct but misses the obvious: that with sediment addition the negative effect of higher velocity disappeared. Framed in this latter way, the next sentence (lines 292-294), which I cannot understand, becomes unnecessary. 

R: This paragraph has been rewritten and the sentence removed. Line 314-320.

Line 297: So how might the sediments have increased the resistance of algae to higher flow? This question needs answering and I suggest the answer lies in the nutrient effect. Horner and Welch (1981) found that the optimum velocity for algal growth increased with increasing phosphorus concentration. Basically, at higher velocities, the higher nutrients allow the algae to grow faster than they are sheared. In your case, the added nutrients may have shifted the optimum upward enough to eliminate the negative velocity effect.

R: Paragraph rewritten. We have follow the reviewer suggestion about optimum velocity for algal growth to discuss our results. Line 314-320.

Line 299 In what sense? I’m confused.

R: Removed. 

Lines 308-313. The higher metabolic efficiency at low flows in the presence sediment, where there is more chlorophyll, cannot be related to self-shading if we take chlorophyll as an indicator of biomass. Under self-shading, the ratio of GPP to biomass goes down, not up. In the absence of sediment, the increase in metabolic efficiency with flow is associated with a loss of chlorophyll (biomass), so in this case, the trend is consistent with self-shading.

R: Paragraph rewritten. We have follow the reviewer comment and we have corrected the metabolic efficiency discussion focusing on self-shading for no-sediments treatment and silt-shading for sediments treatment. Line 324-333.

Reviewer #2: Review: Perez-Calpe et al. interactive effects of flow and fine sediments on stream biofilms

Overall I enjoyed reading this study, and only have one significant recommendation related to how the authors treat the concept of scouring flows in the context of their replicated stream channels. I think the author’s conclusions are valid and support what has previously been published, but am not convinced that there are significant new insights provided. They certainly make a case that the interaction between suspended and deposited sediments and their subsequent contribution to nutrient loading and periphyton metabolism is important to consider. E.g., not all sediment is created equal. They also provide evidence that the way in which sediment is deposited onto the biofilm is important, but it is a bit challenging to apply these results to the ‘real world’ because of some experimental design artifacts. Finally, I DO believe these concerns can be addressed and that the manuscript would be valuable to readers in this subdiscipline.

Abstract

Line 34: Why would lower water velocity lower sediment deposition? Doesn’t make sense to me. Faster water velocity should entrain more sediment and keep the tiles clearer of sediment deposition?

R: Sediment deposition results from the balance between inputs, which increase with discharge, and outputs, which increase with flow velocity (i.e., with discharge). The exact outcome will depend on the details of experimental settings. In our case, flow velocity in all channels was rather slow, so the effect of input differences among channels prevailed over the effect of output differences. We have clarified this in the methods and results sections. 

Paragraph ending with line 78: I really liked this summary of the varied and complex interactions that occur with biofilm metabolism as a result of flow modification. Excellent.

Line 99: change ‘was’ to ‘were’. 

R: It is the bottom that was covered, so we left the text as it was. Line 101.

Line 103: were grazers kept from incubating tiles? Please clarify.

R: It has been clarified in the text. Line 106.

Line 110: Did you measure/quantify dissolved nutrients? Seven days of recycled water could lead to significant nutrient reduction although you may have quantified this.

R: Yes, we measured dissolved nutrients before and after renewing water (line 141). We have added more information about parameters on results section (line 204). 

Line 121: A couple of concerns here: in a ‘natural’ flood, sediment would remain in suspension for a longer period of time and abate during the falling limb of the hydrograph. By not resuspending the sediment additions, I’m not sure how much of an effect you would have, although this may become evident when looking at your AFDM or turbidity data (e.g., suspended sediment was deposited within the stream channels based on water velocity characteristics). Another concern with the sediment addition is related to dissolved and sediment-derived nutrients. As you noted in the introduction, sediment comes hand-in-hand with nutrients that can be released and utilized by the biofilm. So, how do you know that the results you obtained weren’t due to differences in dissolved nutrients between treatments? This is an important, potentially confounding factor in your study design. Perhaps you include this in your analyses.

R: About the first concern, we agree with the reviewer with sediment deposition on natural floods, but in addition to flood-related pulses, turbidity pulses are quite common in our streams, for instance, when forestry machinery crosses over stream channels. We have clarified this in the text. Moreover, in our experiment we focused on the effects of the amount of sediment deposited and not into the effects of turbidity, for this reason we quantified the amount of sediment per unit of surface.

Regarding the second concern, we concluded that the differences between treatments were produced by nutrients derived from sediments, as shown by their leachates. We now describe and discuss this in greater detail. 

Line 191: Interesting! Why do you think conductivity was lower in the sediment treatment? This doesn’t make sense to me unless the sediment was adsorbing dissolved ions. 

R: This is interesting indeed. It is possible that the sediment was adsorbing some dissolved ions. Unfortunately, our data do not allow us to test the potential relevance of this mechanism. Line 199.

Line 214: Table 2. Is there a way to indicate the direction the significant interactions? For example, deposited sediment increased with increasing flow, etc.

R: Following the reviewer suggestion, we have added a new column in Table 2, with the sign of the coefficient when the source of the variation is significant. 

Line 236: Although I agree with your interpretation of the data—namely that sediment deposition on the tiles increased metabolism, I don’t think you can claim that higher flows resulted in scouring. If that were the case, sediment deposition on the tiles would be lower in the high velocity treatments. Right? I think you can fix this by changing the wording here, and would strongly advise that you restructure the materials/methods too. I’d refrain from talking about scouring flows, and instead simply note that sediment deposition in channels with higher flow and velocity was an intentional artifact of your sediment/no-sediment treatment. It seems to me that when you dosed your feeder tanks with sediment, there was more sediment deposited on tiles in channels with higher flows because they were pushing more sediment (mass per unit time) over the channel. Does this make sense? Obviously there is no scouring going on at these velocities or you wouldn’t see this pattern.

R: We have clarified the blocks functioning in the methods section (line 126). Results section has been rewritten (line 223-231), and we replaced “scour” with “slough” in the text. 

Line 244: change ‘showing’ to ‘showed’. 

R: Done. Line 259.

Line 245: I am looking at Figure 3, and although I appreciate your statistical inference, I’m wondering if you believe it is ecologically significant? E.g., is the pattern indicating an ecological response, or is it an artifact of the experimental design and indicative of significant variability in the data?

R: Following the comments of the other reviewer, we detected an error in the figure 3 (now Fig 4). We have change the adjustment lines and the interpretation. We hope answer to his comment. Regarding its ecological relevance, we believe it is, although it might not occur when experimental conditions differ very much from ours.

Line 276 and preceding paragraph: Excellent review of some of the key papers looking at biofilm/velocity relationships.

Line 317: Discussion—Very nice. In particular, I appreciated your thoughtful review of your initial hypotheses. I also appreciated your observation that velocities used in this study were on the low side relative to the literature (aka Biggs et al., etc.).

Minor point: I find that some readers mistake ‘flow’ for ‘velocity’. I’d recommend that you change this to ‘discharge’ throughout. 

R: Following the reviewer suggestion, we have replaced “flow” with “discharge” in all the text.

---

## [Decision Letter · Decision Letter 1]

11 Jan 2021

PONE-D-20-22278R1

Interactive effects of discharge reduction and fine sediments on stream biofilm metabolism

PLOS ONE

Dear Dr. Perez-Calpe,

Thank you for submitting your revised manuscript to PLOS ONE. After careful consideration, we continue to feel that it has merit but still does not fully meet PLOS ONE’s publication criteria as it currently stands. Therefore, we invite you to submit a second revision of the manuscript that addresses the points raised during the review process for the first revision.

Both reviewers note (and I agree) that you did a good job of improving the manuscript. Reviewer #2 is satisfied with your current revision and recommends acceptance. Reviewer #1 has some new concerns with the current revision and recommends a few additional changes (minor revision) before acceptance. I agree with Reviewer #1 and ask that you make the relatively simple but still important revisions he recommends (I highlight these in the next paragraph). Both reviewers also suggest some minor editorial changes that should help clarify a few passages in your manuscript, so please consider making these changes, as well.

The main substantive comments that Reviewer #1 makes are as follows. He urges you to restore the solid and dashed lines that were included in Fig. 1 of your original submission but are missing from the figure (now Fig. 2) in the revised manuscript. Please read his rationale carefully and note the three points he makes in connection with this figure and its role in your manuscript. Please also think carefully about the reviewer's comments on your presentation of results on metabolic efficiency and, in particular, line 216 of the revised manuscript. Additionally, note his question about lines 139--140 (dealing with forms of nitrogen other than nitrate and ammonium), his suggestion regarding line 140 (he suggests expressing N and P as concentrations instead of masses), his comment on Fig. 3 (he correctly notes that it is standard practice in studies of stream metabolism to present GPP in dimensions of oxygen mass per volume per time or mass per horizontal area per time rather than in dimensions of mass per time, as you do in Fig. 3), and his concern and suggested solution regarding lines 332--333. In my opinion, all of these concerns are valid, so please think carefully about them and make an appropriate revision in each case or explain clearly why you think the revision is not necessary.

We look forward to receiving your revised manuscript.

Kind regards,

James N. McNair, Ph.D.

Academic Editor

PLOS ONE

Reviewers' comments:

Reviewer's Responses to Questions

**Comments to the Author**

1. If the authors have adequately addressed your comments raised in a previous round of review and you feel that this manuscript is now acceptable for publication, you may indicate that here to bypass the “Comments to the Author” section, enter your conflict of interest statement in the “Confidential to Editor” section, and submit your "Accept" recommendation.

Reviewer #1: (No Response)

Reviewer #2: All comments have been addressed

2. Is the manuscript technically sound, and do the data support the conclusions?

Reviewer #1: Partly

Reviewer #2: Yes

3. Has the statistical analysis been performed appropriately and rigorously? 

Reviewer #1: No

Reviewer #2: Yes

4. Have the authors made all data underlying the findings in their manuscript fully available?

Reviewer #1: Yes

Reviewer #2: Yes

5. Is the manuscript presented in an intelligible fashion and written in standard English?

Reviewer #1: Yes

Reviewer #2: Yes

6. Review Comments to the Author

Reviewer #1: Reviewer #1

The paper is greatly improved from the first draft. As a referee I found it highly gratifying to have essentially all of my comments taken seriously and responded to in a substantive way. There do remain a few issues, which are noted below along with minor editorial notes.

I have a new concern, however, which is that the lines in Fig. 2 (formerly Fig. 1) should not have been removed and should be restored. I agree with the removal of the R-squared values but I think it essential to show the solid and dotted lines together with an explanation in the legend that that separate lines imply a significant interaction. The reason is that the mixed-model (LMEM) results provided in Table 2 do not tell the whole story. The significance of the main effect of a covariate says nothing definitive about the significance of the individual within-treatment slopes. A significant interaction says only that the slopes are different.

In the case of effects on chl-a (where the problem is most serious), the important Points were: (1) in the absence of sediments, discharge had a negative effect, (2) sediments had a positive effect; and (3) in the presence of sediments, the negative effect of discharge vanished. Table 2 tells us only that sediments have a significant main effect and that that there is an interaction with discharge. The text, as a result, appears contradictory, stating first that the no-sediment chl-a declined with discharge (but without a clear statement that this decline was significant), then that the effect of discharge was not significant. It references the interaction but fails to specify that this interaction reflected two slopes, one significantly negative (no-sediments) and the other essentially zero (sediments). Point (3), therefore is lost altogether. Fig.2C, as it stands, is of little help—just a bunch of points that are difficult to interpret. The lines (as on old Fig 1 C) make all the difference. They make the trends easy to see and, more importantly, they provide the reference necessary to report all three Points in the text. In the original submission (with the lines), all three Points were presented in the text. Point (3) must be presented as a Result because it is referenced in the Abstract (line 38) and is addressed in the Discussion (line 316 and following).

In the case of deposited sediment (Fig. 2B), the issue is less important, but the text nonetheless suffers from the absence of the within-treatment slopes. If the slopes were available for reference it would be possible to correct the sentence on lines 226-229 to clarify that the discharge affected sediment deposition, but only in the sediment treatment; and to strike the unhelpful last sentence (line 229-231) of the paragraph. (I am afraid that it was my comment to Line 210 of the original submission--“Did deposited sediment really increase with flow in the no-sediment treatment?”—that led to the removal of the trend lines from Fig. 2 and for that I apologize. I didn’t mean to challenge the lines, just the misinterpretation of the Table-2 stats.)

In the case of metabolic efficiency (Fig. 4) you have retained the slopes (now corrected), which I applaud. Here, they are obviously essential to the presentation. However, the text remains problematic. Although you didn’t see main effects, it seems misleading to say that there was “no significant effect of discharge or sediments” because in the no-sediments treatment there was a clear effect of discharge on efficiency. SAS Proc Mixed run on your dataset shows this effect (the slope of this line) to be significant. I also get that the slightly negative slope for sediments treatment was not significant. For this reason, I would change the phrase on line 216 which reads “in the presence of sediments, increasing discharge resulted in a lower metabolic efficiency” (not significant) to “in the absence of sediments, increasing discharge resulted in a higher metabolic efficiency” (significant).

In the above I have skirted the question of whether the values of the slopes and their significance probabilities should be reported. My sense is that it would be sufficient to report in the text or figure legends simply whether a given slope is or is not significant. I am shaky ground here and am not familiar with lme in R, but I find using SAS Proc Mixed that, where there is a significant interaction, one needs to remove the main effect (discharge in this case) from the model statement to get the within-treatment slopes and significance, which otherwise are reported relative to the respective main-effect values.

Other comments:

Keywords: spell out chlorophyll-a

36: hyphenate: “no-sediments”

37-38: If there is room, acknowledge that nutrients may explain the sediment effect.

107: The schematic of Fig. 1 helps greatly. Thank you.

132: I suggest “Sediments leachate” should be “Sediment leachate.” Consider making this change throughout, wherever “sediments” is used as an adjective.

139: strike “with”

139-140. I see a discrepancy here. Are you counting only the ammonium and nitrate to get the 20.9 mg added? If so, please clarify. The phosphorus looks right.

140: I think the quantities of N and P would be better expressed as concentrations, i.e., by dividing by the block water volume if I understand correctly.

140: “to the each” should read “to each”

206-209. I would not call the changes in SRP, NO3, and DOC “small”. The increase in SRP, for example, was about equal to the average concentration. It may be useful to note that most of the increases appeared to come from the new water, rather than from the sediments.

261: As noted above, should read: “in the absence of sediments, increasing discharge resulted in a higher metabolic efficiency”.

263: Fig. 3. Need to express oxygen in units per area.

316: “consisted on” should be “consisted of”.

317: Change “shows…conditions” to “is consistent with a fertilizing effect that is stronger than the sloughing effect” because the fertilizing effect is only suggested, not established, and the response doesn’t show anything about the fertilizer effect if it doesn’t exist.

321: Baatrup et al. should be “Baattrup-Pedersen et al.” I hope you are right on the point made in this sentence; I could not understand that paper.

332-333: This still needs work. We might expect sediment shading to reduce metabolic efficiency, but in your case, the metabolic efficiency was NOT reduced; it was the same as no-sediments at high flows and higher than no-sediments at low discharge. My explanation: Sediment had a greater stimulatory effect on GPP than on chl a . At low flows this more than compensated for any sediment-shading effect, so sediment efficiency was greater than no-sediment efficiency. At high flows the stimulatory effect of the sediments was matched by the enhancing effect of the reduced self-shading in the no-sediment treatment, so the efficiencies were about equal.

Reviewer #2: Perez-Calpe et al. Interactive effects of Q reduction and fine sediments on stream biofilm metabolism.

I appreciate the author’s due-diligence responding to reviewer suggestions and questions and the manuscript is improved. I particularly appreciate the clarification presented in Table 2, coupled with Figures 3 and 4 and think this makes for a compelling argument for the interactive effects of sediment-derived nutrient addition and biofilm stimulation (e.g., bulk rates of GPP and NCM) vs. photosynthetic efficiency (GPP/chl-a ratio). Figure 1 is also a helpful addition.

Specific comments:

Line 139: Please recast for clarity. Delete ‘with’ and ‘the’ (line 140).

Line 302-305: Clarify how you quantified ‘type of biofilm’. This is confusing as you state at the beginning of the sentence that you did not study algal composition.

7. PLOS authors have the option to publish the peer review history of their article (what does this mean?). If published, this will include your full peer review and any attached files.

Reviewer #1: No

Reviewer #2: **Yes: **Eric Snyder

---

## [Author Response · Author response to Decision Letter 1]

18 Jan 2021

Dear Dr. McNair,

A new revised version of the manuscript PONE-D-20-22278R1 “Interactive effects of discharge reduction and fine sediments on stream biofilm metabolism” by Ana Victoria Pérez-Calpe, Aitor Larrañaga, Daniel von Schiller and Arturo Elosegi, is attached to your consideration for publishing as an article in PLOS ONE journal. 

We are very grateful again to the reviewers for their detailed and positive comments. We have followed closely these comments, as we detail below. 

We hope that the present version of the Ms reaches the standards for PLOS ONE journal.

Best regards,

Ana Victoria Pérez-Calpe, PhD student

Faculty of Science and Technology

University of the Basque Country,

P.O. Box 644, 48080 Bilbao, Spain

Tel.: +34 94 601 5939

Email: anavictoria.perez@ehu.eus

--------

Reviewer #1

The paper is greatly improved from the first draft. As a referee I found it highly gratifying to have essentially all of my comments taken seriously and responded to in a substantive way. There do remain a few issues, which are noted below along with minor editorial notes.

I have a new concern, however, which is that the lines in Fig. 2 (formerly Fig. 1) should not have been removed and should be restored. I agree with the removal of the R-squared values but I think it essential to show the solid and dotted lines together with an explanation in the legend that that separate lines imply a significant interaction. The reason is that the mixed-model (LMEM) results provided in Table 2 do not tell the whole story. The significance of the main effect of a covariate says nothing definitive about the significance of the individual within-treatment slopes. A significant interaction says only that the slopes are different.

R: We have restored all the trend lines in Fig 2 and we have added the explanation in the legend.

In the case of effects on chl-a (where the problem is most serious), the important Points were: (1) in the absence of sediments, discharge had a negative effect, (2) sediments had a positive effect; and (3) in the presence of sediments, the negative effect of discharge vanished. Table 2 tells us only that sediments have a significant main effect and that that there is an interaction with discharge. The text, as a result, appears contradictory, stating first that the no-sediment chl-a declined with discharge (but without a clear statement that this decline was significant), then that the effect of discharge was not significant. It references the interaction but fails to specify that this interaction reflected two slopes, one significantly negative (no-sediments) and the other essentially zero (sediments). Point (3), therefore is lost altogether. Fig.2C, as it stands, is of little help—just a bunch of points that are difficult to interpret. The lines (as on old Fig 1 C) make all the difference. They make the trends easy to see and, more importantly, they provide the reference necessary to report all three Points in the text. In the original submission (with the lines), all three Points were presented in the text. Point (3) must be presented as a Result because it is referenced in the Abstract (line 38) and is addressed in the Discussion (line 316 and following).

R: We have changed the text to avoid contradictions and to present the third point. 

In the case of deposited sediment (Fig. 2B), the issue is less important, but the text nonetheless suffers from the absence of the within-treatment slopes. If the slopes were available for reference it would be possible to correct the sentence on lines 226-229 to clarify that the discharge affected sediment deposition, but only in the sediment treatment; and to strike the unhelpful last sentence (line 229-231) of the paragraph. (I am afraid that it was my comment to Line 210 of the original submission--“Did deposited sediment really increase with flow in the no-sediment treatment?”—that led to the removal of the trend lines from Fig. 2 and for that I apologize. I didn’t mean to challenge the lines, just the misinterpretation of the Table-2 stats.)

R: Done

In the case of metabolic efficiency (Fig. 4) you have retained the slopes (now corrected), which I applaud. Here, they are obviously essential to the presentation. However, the text remains problematic. Although you didn’t see main effects, it seems misleading to say that there was “no significant effect of discharge or sediments” because in the no-sediments treatment there was a clear effect of discharge on efficiency. SAS Proc Mixed run on your dataset shows this effect (the slope of this line) to be significant. I also get that the slightly negative slope for sediments treatment was not significant. For this reason, I would change the phrase on line 216 which reads “in the presence of sediments, increasing discharge resulted in a lower metabolic efficiency” (not significant) to “in the absence of sediments, increasing discharge resulted in a higher metabolic efficiency” (significant).

R: Changed.

In the above I have skirted the question of whether the values of the slopes and their significance probabilities should be reported. My sense is that it would be sufficient to report in the text or figure legends simply whether a given slope is or is not significant. I am shaky ground here and am not familiar with lme in R, but I find using SAS Proc Mixed that, where there is a significant interaction, one needs to remove the main effect (discharge in this case) from the model statement to get the within-treatment slopes and significance, which otherwise are reported relative to the respective main-effect values.

R: We understand the concern of the referee here: significant interaction means crossing lines and, thus, the interpretation is linked to the range of the covariate of the analysis. We have interpreted the results based on the models and the figures. Even with a significant interaction term, if we were observing a significant effect of our treatment AND the figure was showing evidence that values in our discharge range were different comparing one treatment with the other, we have stated that difference.

Other comments:

Keywords: spell out chlorophyll-a

R: Done 

36: hyphenate: “no-sediments”

R: Done

37-38: If there is room, acknowledge that nutrients may explain the sediment effect.

R: Done

107: The schematic of Fig. 1 helps greatly. Thank you.

R: Thanks

132: I suggest “Sediments leachate” should be “Sediment leachate.” Consider making this change throughout, wherever “sediments” is used as an adjective.

R: Done

139: strike “with”

R: Done

139-140. I see a discrepancy here. Are you counting only the ammonium and nitrate to get the 20.9 mg added? If so, please clarify. The phosphorus looks right.

R: Yes, this value is the result of sum ammonium and nitrate.

140: I think the quantities of N and P would be better expressed as concentrations, i.e., by dividing by the block water volume if I understand correctly.

R: As the water is recirculating, for some purposes it may be more interesting to express these quantities as total mass. Therefore, we express them in both ways: mass and concentration.

140: “to the each” should read “to each”

R: Done

206-209. I would not call the changes in SRP, NO3, and DOC “small”. The increase in SRP, for example, was about equal to the average concentration. It may be useful to note that most of the increases appeared to come from the new water, rather than from the sediments.

R: Clarified.

261: As noted above, should read: “in the absence of sediments, increasing discharge resulted in a higher metabolic efficiency”.

R: Done

263: Fig. 3. Need to express oxygen in units per area.

R: Changed and S1. Dataset has been corrected too. 

316: “consisted on” should be “consisted of”.

R: Done

317: Change “shows…conditions” to “is consistent with a fertilizing effect that is stronger than the sloughing effect” because the fertilizing effect is only suggested, not established, and the response doesn’t show anything about the fertilizer effect if it doesn’t exist.

R: Changed

321: Baatrup et al. should be “Baattrup-Pedersen et al.” I hope you are right on the point made in this sentence; I could not understand that paper.

R: We have corrected the reference. We also have checked the paper and the sentence about it is correct.

332-333: This still needs work. We might expect sediment shading to reduce metabolic efficiency, but in your case, the metabolic efficiency was NOT reduced; it was the same as no-sediments at high flows and higher than no-sediments at low discharge. My explanation: Sediment had a greater stimulatory effect on GPP than on chl a . At low flows this more than compensated for any sediment-shading effect, so sediment efficiency was greater than no-sediment efficiency. At high flows the stimulatory effect of the sediments was matched by the enhancing effect of the reduced self-shading in the no-sediment treatment, so the efficiencies were about equal.

R: Modified. 

Reviewer #2: 

I appreciate the author’s due-diligence responding to reviewer suggestions and questions and the manuscript is improved. I particularly appreciate the clarification presented in Table 2, coupled with Figures 3 and 4 and think this makes for a compelling argument for the interactive effects of sediment-derived nutrient addition and biofilm stimulation (e.g., bulk rates of GPP and NCM) vs. photosynthetic efficiency (GPP/chl-a ratio). Figure 1 is also a helpful addition.

Specific comments:

Line 139: Please recast for clarity. Delete ‘with’ and ‘the’ (line 140).

R: Done

Line 302-305: Clarify how you quantified ‘type of biofilm’. This is confusing as you state at the beginning of the sentence that you did not study algal composition.

R: Changed.

---

## [Editor Report · Decision Letter 2]

26 Jan 2021

Interactive effects of discharge reduction and fine sediments on stream biofilm metabolism

PONE-D-20-22278R2

Dear Proto-Dr. Perez-Calpe,

We’re pleased to inform you that your manuscript has been judged scientifically suitable for publication and will be formally accepted for publication once it meets all outstanding technical requirements.

Kind regards,

James N. McNair, Ph.D.

Academic Editor

PLOS ONE
---

## [Editor Report · Acceptance letter]

28 Jan 2021

PONE-D-20-22278R2 

Interactive effects of discharge reduction and fine sediments on stream biofilm metabolism 

Dear Dr. Perez-Calpe:

I'm pleased to inform you that your manuscript has been deemed suitable for publication in PLOS ONE. Congratulations! Your manuscript is now with our production department. 

Kind regards, 

on behalf of

Dr. James N. McNair 

Academic Editor

PLOS ONE